# Urban Public Policy and the Formation of Dilapidated Abandoned Buildings in Historic Cities: Causes, Impacts and Recommendations

**Hamed Tavakoli [1] and Massoomeh Hedayati Marzbali [2,\*]**

[1]  Faculty of Architecture, Building and Planning, The University of Melbourne, Masson Rd, Parkville, VIC 3010, Australia; hamed.tavakoli@mail.com

[2]  School of Housing, Building and Building, Universiti Sains Malaysia, Gelugor 11800, Penang, Malaysia

\*  Correspondence: hedayati@usm.my; Tel.: +60-17-447-1295

**Abstract:** The contradictory and inefficient nature of urban public policy in Iranian historic cities has been subject to long debates in recent years, and has even led to disorganisation in the formation of dilapidated abandoned buildings (DABs). Under the current policies, three government agencies oversee the urban management in historic cities. The projects and processes that have yet to be implemented by these agencies are crucial for solving the problems associated with DABs. This research aims to investigate the current public policies of the essential key players and stakeholders in order to ameliorate the problem of DABs, which in the literature has been proven to be associated with socio-spatial disadvantage. A qualitative semi-structured enquiry was conducted, and urban public policies were evaluated on the basis of 19 in-depth interviews regarding the three historic cities of Yazd, Kashan and Isfahan. The results were analysed using cutting and sorting techniques, and thematic and critical narrative analysis. Several inadequacies in the current urban public policy were specified. This research could help decision-makers to create efficient management plans with respect to the reduction of DABs, an approach that can be considered efficient for the regeneration of life in historic cities.

**Keywords:** urban public policy; historic Iranian cities; dilapidated abandoned buildings; original residents

## 1. Introduction

After hundreds of years of morphological consistency and organic growth, present-day Iranian cities have become subject to an unprecedented phenomenon that initially occurred at the beginning of the 20th century. The new social and economic changes eventually generated poverty, unemployment and inequality in the access to public services and infrastructure amongst the residents of historic urban areas, culminating in a swift urban sprawl both outside and on the fringe of historic cities [1,2]. The urban transformations imposed by modernity demolished the old city walls and dramatically changed the physical spatial configurations of old cities [3]. According to researchers of contemporary Iranian urbanism, the exogenous socio-spatial movements since the 1920s have reshaped the historic cities [2]. Traditional cities have been carved out and transformed under capitalism and modernity to accommodate vehicular access and modern urban functionalities (Figure 1). Present-day traditional commercial structures have lost their consistency and significance as a result of the ever-rising demand for modern modalities (e.g., vehicular accessibility), whereas contemporary city fringe developments have expanded outwards, stretching far beyond the historic centres [3].

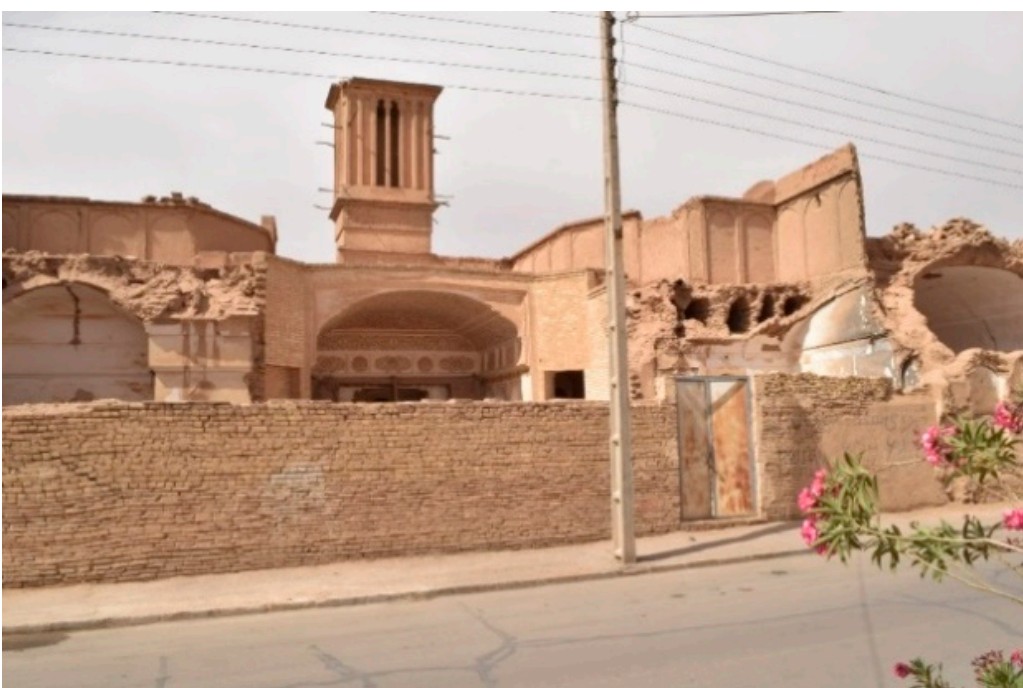

**Figure 1.** Historic cities are carved by new road developments which have turned traditional fabrics into disaggregated urban tissues and generated large areas of DABs (Yazd, Iran 2018) (source: the authors).

Thousands of historic monuments in Iran, situated in the centres of ancient cities, are registered as national heritage sites. Most of these monuments are great examples of traditional Iranian architecture and urban design, which can be regarded as important foundations of the tourism industry in Iran. However, many of these large monuments are deeply embedded in the narrow urban fabrics. As a result, government agencies may not be able to facilitate and establish contemporary urban infrastructures in close proximity to such dense built environments.

Historic urban fabrics in Iranian cities have mainly been subject to gradual decay, with an exodus of the population and the abandonment of buildings for more than half a century [4–6]. Such rapid socio-spatial transformation has generated multiple urban issues. In the backdrop of these chronic socio-spatial problems, researchers have endeavoured to define several urban public policies/incentives to address dilapidated abandoned buildings (DABs) in the Iranian planning context, especially because the existing efforts have been proven ineffective [7]. Consequently, this study aims to delve into some of the less-discussed issues which are relevant to the current public policies in historic Iranian cities. With this objective, the study conducts qualitative research by implementing in-depth semi-structured interviews with policymakers and practitioners in three historic cities of Iran.

*1.1. Geographical Context and Differences in Inner City Transformation in Historic Middle Eastern and European Cities*

Since 1920, modernisation strategies and urban development trends in Iran and Middle Eastern countries have justified the spatial transformation, redevelopment, demolition and destruction of traditional urban fabrics as a method of providing contemporary requirements for the residents. Nevertheless, the disagreement over the value of Iranian urban cores and the inevitable modification of urban areas has created a problematic condition regarding the protection of the historic environment, specifically when the issue of revitalisation is equated with those of European counterparts [8].

By looking at European cultural history, one can claim that the urban transformation of historic cities in Europe started with the Renaissance in the 14th century and continued to the Enlightenment, culminating with the 19th and 20th century industrial movements [9]. Therefore, the whole process of urban transformation, including the adoption of modernity by Western cities, materialised in almost 500 years. This transition facilitated a reasonable time duration for the socio-spatial integration of historic cities with the surrounding modern built environments. By contrast, the whole process of modernisation in Middle Eastern cities, which was launched at the beginning of the 20th century, transformed traditional structures in only a few decades. The corresponding geographical historic differences can be seen as the central reason why Western models of urban renewal cannot yield an acceptable level of outcome amongst Iranian or Middle Eastern historic cities [10].

By observing the history of urban transformation, one can see that Western intervention has revolutionised the architectural types from single-building restoration models (in the early 19th century) to a regeneration in the 2000s and beyond. Hence, contemporary models of Western intervention in historic areas are largely based on comprehensive planning, public participation, mixed restoration−gentrification, the adaptive reuse of heritage buildings, changes in land use, the provision of financial motivation and the implementation of mixed-use buildings. These approaches can reasonably regenerate old cities in Western culture [11]. In particular, the Western methods of urban renewal have viewed old cities as large-scale urban museums within historic areas that have adapted to the modern necessities of life over five centuries of consistent structural modification [10].

By contrast, the rapid and unfiltered adaptation of 'modernity' in Iranian or Middle Eastern historic cities devaluated the land and properties in the heritage nucleus, generated massive socio-spatial degradation and assisted in rural immigration towards the old urban context [10]. Such socio-spatial transformation was consequently followed by superimposing the new city onto old historic fabrics and cutting out wide new roads for vehicular accessibility and infrastructure [3]. The latter entailed a progressive demolition of historic fabrics, which has continued until the present day. These socio-spatial qualities are expected to cause a dangerous rift between the new elite who live in newly emerging suburbs and the ordinary people who inhabit the old city [12].

## 2. Research Context: DABs as a Chronic Problem in Historic Cities

All natural structures and human built-environments are subject to constant deterioration, indicating that urban decay can be viewed as an indispensable part of urban life, possibly occurring at different paces and for various reasons [13]. Urban deterioration is a result of interrelated socioeconomic conditions that may be caused by deindustrialisation, lowered land value, economic breakdown and the failure of businesses, which lead to increased crime rates, growing unemployment and rising poverty. These conditions are evident in abandoned buildings, overrun sewers, street areas with trash and rubble, and deserted landscapes [14,15]. Another compelling reason of urban decay is the socioeconomic development of nearby areas to which the population has migrated for better opportunities [16,17], a scenario that differs from the historic cities in Iran [18]. Urban deterioration has a deleterious effect on historic cities because it creates a disorganisation imbalance, a decline in socio-spatial characteristics, illegibility, a lack of vehicular accessibility and a shortage of socio-physical urban infrastructures, which subsequently eradicate socio-spatial memories [19]. Three types of socio-spatial scenarios may be evident. Firstly, socio-functional deterioration accompanies unflawed spaces. Secondly, physical deterioration is accompanied by vigorous socio-spatial functionality. Finally, both social and spatial deterioration can occur simultaneously [20].

In response to the deleterious effects of urban decay in historic cities worldwide since the 18th century, several global movements, such as the Society of Antiquaries of London, have reiterated the need to revitalise heritage sites and cultural properties [21]. Since the 1970s, historic cities have undergone a reassessment of their importance. In the 21st century, the historic revitalisation has been largely associated with city planning and development.

Advocates have promoted preservation as a key driver of urban revitalisation; however, the empirical research addressing this connection is limited [22].

The revitalisation of historic Middle Eastern cities was initiated towards the end of the 1960s for two reasons. Firstly, thousands of rural migrants were trickling into the city, where the medina was the obvious and only place to find cheap accommodation. Secondly, the rapid pace of urbanisation transformed the historic Middle Eastern city into a hollow space. For many years, the cultural heritage character of urban fabrics, as widely recognised by UNESCO, called for the regeneration of Islamic cities [23].

### 2.1. DABs in Historic Iranian Cities

As a result of the unprecedented contemporary urban transformation of traditional cities, the present-day large areas of historic fabrics can be considered to be areas of DABs, albeit that some disused areas have existed for six decades [24]. Mirmiran [18] (p. 63) suggested that 12.7% of all historic areas in Kashan are composed of DABs. Behzadfar [25] (p. 73) reported that approximately 15% of urban areas inside the historic fabrics in Yazd can be classified as DABs, which attract antisocial behaviours, are places of poor communities and generate a perceived or actual lack of safety. Thus, present-day historic Iranian cities have transformed into fragmentary fabrics, either remaining unattended for decades or being replaced by new developments that do not have harmonious relationships with their surrounding environs. DABs and the relevant redevelopment regulations can be considered a challenging concept because they have been largely neglected in the context of socio-spatial planning [7].

### 2.2. DABs and Socio-Spatial Disadvantage

In historic Iranian cities, the presence of disadvantaged (non-local) economic migrants and foreign refugees is noticeably correlated to high ratios of DABs and building deterioration. For those families whose lives are unstable in the diaspora, seeking sanctuary in minimal living facilities inside historic zones seems to be a good option, as the setup can be quite tolerable to them, either in contrast to their original life in the villages or their severe poverty and homelessness [18]. The lack of vehicular accessibility can also be viewed as a major factor of the emigration of the original residents, subsequently facilitating an influx of refugees and concomitant antisocial behaviours inside the historic city, which in turn have yield socio-spatial deterioration [18]. The unprecedented transformation has exceedingly undermined the social capital in such areas, and current residents seriously doubt whether authorities can generate effective change—a phenomenon that can create additional DABs [26]. At present, due to the narrow, winding streets, the areas inside historic urban areas often cannot accommodate modern infrastructure, which consequently has deprived communities of essential public services [27] and encouraged the original residents to leave these historic cities [28]. The corresponding form and spatial arrangements of traditional houses also do not respond to modern human needs, and such a lack of spatial responsiveness has created more DABs [29].

### 2.3. Lack of a Sense of Belonging to a Place and DABs

The lack of a sense of belonging to a place amongst residents is the result of the lack of a sense of place-satisfaction, which may be generated by socio-spatial deterioration, exposure to widespread DABs or a lack of public infrastructure [30]. Contemporary cities have jeopardised the structures of public spaces in traditionally built environs, whilst modern streets and vehicular roads are inharmonious to the network of public space and urban life in the old cities. These spatial conflicts between old and new structures, which are widely distributed because of the demands of modern urban zoning and vehicle traffic, have led to the generation of inhumane spaces that further discourage the human presence in historic areas [31]. In many circumstances, the absence of social safety, the existence of hygienic and cultural problems, and the lack of infrastructure and social capital may have also contributed to the low sense of belongingness, consequently leading the original

residents to leave the heritage cores [32]. Obtaining cheaper housing opportunities can also be regarded as a major reason for the immigration of refugees and non-local disadvantaged communities towards the historic cities [33]. Moreover, the presence of non-local residents who are unfamiliar with the heritage values of the affected urban communities can further contribute to the debilitation of historic cores and even worsen the sense of belonging of all residents [28].

### 2.4. Identity Crisis and DABs

The loss of a sense of place-identity amongst residents is widely reported in Iranian historic cities [34], an issue that is relevant to the disinterest of residents towards their inherited family houses or traditional inhabitants [30]. Nowadays, the lack of a sense of place-identity can be directly related to rapid socio-spatial transformation, which has coincided with the initiation of contemporary lifestyles [35–37]. Thus, the changing trends and technological development can be perceived as significant factors that have damaged the sense of identity amongst residents in historic cities [38]. Within the historic fabrics, meaningful grounds can help to depict the strong correlation amongst physical–environmental qualities, levels of social capital and the sense of place-identity amongst residents [39]. Such scenarios can help to establish an assumption regarding the severe identity crisis and the consequent lack of a sense of belonging to the place, which have resulted from the influx of socioeconomic migrants and the formation of DABs (Figure 2).

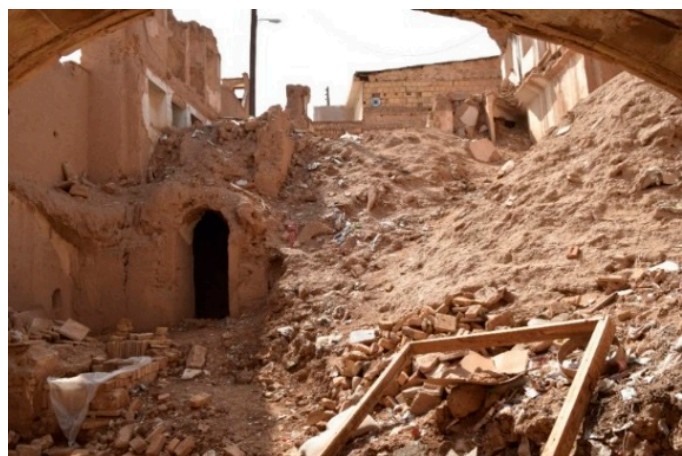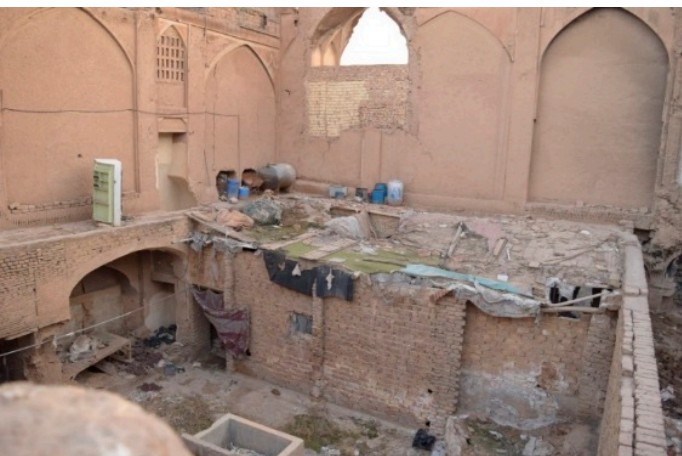

**Figure 2.** DABs and the accumulation of disadvantaged communities have become a serious socio-spatial problem in historic Kashan (left) and Yazd (right) in Iran 2018 (source: the authors).

### 3. Public Policies in Historic Iranian Cities

In Iranian historic cities, three major government agencies oversee the regulation/ management of heritage districts. The first government agency is the Iranian Cultural Heritage, Handicrafts and Tourism Organization (ICHHTO), which aims to provide and implement programmes for the protection, preservation, restoration and revitalisation of cultural–historic sites and urban contexts in Iran [40]. The ICHHTO also plays a local and central role in generating regulations and managing historic urban areas. The organisation is also responsible for providing feedback about strategic plans and detailed development/master plans, as occasionally proposed by the two other government agencies [5].

The second government agency is the Office for Urban Renewals and Improvements (the executive arm of the Local City Council), which acts under the auspices of the Ministry of State, oversees the development of innovative policies and makes decisions on the management of historic sites and old cities. The office's tasks include collaborating with NGOs and crafting identity generation and development plans for a city by rehabilitating, reconstructing and renovating urban fabrics, including historic zones [41].

The third government agency is the Urban Development and Improvement Company, which was established under the auspices of the Ministry for Roads and Urban Development, and acts as a specialised holding corporation. The company aims to directly improve the knowledge of urban management and the renewal, development and promotion of intellectual properties [5]. Its other aims are to consolidate passive policies/programmes and community empowerment schemes, and to observe active executive projects, subsequently providing a stimulating approach towards urban regeneration in historic areas.

Similar to the ICHHTO, the other two government agencies can implement regulations and projects inside historic urban areas. Therefore, proposals can be highly influenced by organisational perspectives, although they hardly correspond with one another [7]. In practice, the traditional urban fabrics, as defined by the ICHHTO, seem to noticeably characterise the deteriorated urban fabrics defined by the other two urban agencies. Therefore, facilitating equally legible urban public policies has become a crucial endeavour [5].

*DABs and Governmental Deficiencies*

The current Comprehensive Planning paradigm, as proposed by the relevant agencies, has failed to resolve the urban development dilemmas of Iranian cities because it largely emphasises the need for integrated planning [42]. Moreover, contemporary urban planning and management suffer from several defects, such as the lack of stakeholder participation, the existence of diverse agencies in the process of urban management, the lack of finance, and the poor implementation of projects and regulations [43,44]. Insufficient vehicular access, a lack of car parking space and a lack of infrastructure have altogether transformed historic cities into problematic districts. The existence of organic geometry, the narrow dimensions of passageways, the lack of infrastructures necessary for modern urban life and the lack of visual integrity between new constructions and old ones have also altogether forced many local residents to leave the historic areas, which then amplifies the formation of DABs.

At present, the rehabilitation of historic cities in Iran and Middle Eastern countries is generally disregarded as a priority objective amongst the relevant government agencies. This inattention may be attributed either to the obsolete image of historic areas amongst the public or the lack of technical and/or institutional capability (and capacity) to comprehend the complexity of the physical and social rehabilitation problems [45]. Hence, regardless of whether the issue is wholesale demolition or the widespread neglect of DABs, the common problem is that most decision-makers identify themselves with a development process that is alien to the cultural traditions of their respective society. Moreover, decision-makers are rarely provided with suitable technical approaches and institutional tools that can truthfully demonstrate the viability of the alternatives and appropriate models of intervention [10].

Historic urban cores in Iranian cities are largely undermined in the various development strategies. For instance, the underlying emphasis is on Western physical–linear regeneration, and the delivery of flagship projects is considered the prevalent approach; both of them are mainly employed by the central government [46]. Nonetheless, the interventions conducted within these physical frameworks have further exacerbated the existing problems. Furthermore, the current preventative building controls have discouraged building investment in the historic areas, a process that has led to the further devaluation of the land, which may yield more DABs and deteriorated fabrics [47]. Therefore, the current public policy needs to be replaced with innovative and participatory planning approaches [7].

The expensive rates for restoration have forced owners to either abandon or rent their houses as half-destroyed properties, and these concerns may have even further generated socio-spatial marginality [20]. The deleterious socio-spatial effects indicate a state of abnormality that can affect larger areas within historic cities [29]. Hence, a gap in the knowledge can be identified, considering that the relationship between DABs and urban public policies in Iranian historic cities has rarely been considered to be a key component to shape sustainable communities. This research presents an approach for evaluating and

recalibrating the relevant policies. In response to the proposed research project, the two following questions are forwarded: How has the current public policy contributed to the formation of DABs? How can these policies be recalibrated to reduce the number of DABs in the future?

## 4. Methods

The scope of the research in the current study was restricted to the collection and analysis of several datasets pertaining to the three historic Iranian cities of Kashan, Yazd and Isfahan. The research implemented a qualitative approach, scrutinising several instrumental case studies through 19 in-depth interviews. The research involved the identification and classification of quotes or expressions that seemed to be important and relevant, the findings of which formed the backbone of this study. All of the potential interviewees were contacted and briefed prior to the interview via phone, email, or directly in person. A participant information sheet was provided to ensure that the potential participants would have sufficient information to make an informed decision on whether to participate in this research or not.

The interviews lasted for 40 to 60 min and were semi-structured, in order to be able to address several pre-identified topics. The semi-structured interviews were conducted amongst a variety of stakeholders and generated open discussions with representatives from the three government agencies, as well as practitioners, scholars and planners regarding the urban public policies in Yazd, Kashan and Isfahan. The abbreviated names, affiliations and roles of the interviewees are decoded in Appendix B. All of the interviews were conducted in the Persian language, and were audio-recorded and subsequently transcribed verbatim. The age range of the interviewees was between 35 and 65 years old.

The interviewees were asked to explain the aftermath of contemporary planning contexts in the historic areas by addressing several questions pertaining to the formation of DABs. Two sets of questions were prepared (Appendix C). The first set targeted representatives from government agencies (policymakers), whilst the second set addressed professionals and practitioners. The interviewees are cited in this article using a specific coding system. The in-depth interviews enabled the collation of multi-factorial information.

### *Analytical Techniques*

For the information analysis, the current research implemented a qualitative cutting and sorting procedure (i.e., identifying quotes or expressions that seem to be important) and arranging them into similar themes [48]. NVivo 11 software was used for the sorting, coding and thematic analysis. The results of the thematic comparison were analysed in the succeeding level through critical narrative analysis, which culminated in the generation of discussions and recommendations [49]. This research required interactions with human participants. Thus, ethics approval was sought, and was subsequently interpreted as "involving no more than low risk for research participants" by the Office of Research Ethics, Compliance and Integrity, University of Adelaide (Appendix A).

## 5. Results

Three significant themes were developed from the analytical techniques, as summarised in Table 1.

**Table 1.** Domain framework, themes and subthemes based on the thematic analysis.

| Domain Framework | Theme | Subtheme |
|---|---|---|
| Cause and effect of DABs in historic Iranian cities | Government Agencies and Current Public Policies | Department for Roads and Urban Development<br>Local municipalities<br>Iran Cultural Heritage, Handcraft and Tourism Organization (ICHHTO) |
| | City Fringe Developments Versus Historic Urban Areas | The inefficiency of current public policies<br>Governance ambiguity and formation of DABs<br>Lack of public services and the morphology of historic cities<br>Emigration of original residents and immigration of low-income communities<br>Lack of cultural awareness and formation of DABs<br>Generational change and the redevelopment of DABs |
| | Negative Aspects of Contemporary Practices | Different organisational perspectives<br>Lack of assessment tools<br>Plans are physical and do not penetrate deep inside traditional fabrics<br>Agendas are non-holistic and not incorporated in the broader context<br>Displacement of residents by urban public projects |

### 5.1. Government Agencies and Current Public Policies

The sixth strategic plan for the development of Iran generally emphasises the regeneration of dysfunctional urban fabrics inside cities. Thus far, the relevant policies or legal documents which may directly/indirectly target DABs at the national level include: (1) the "national strategic plan for the rehabilitation, improvement, reconstruction and empowerment of deteriorated-dysfunctional urban fabrics" (approved in 2014); (2) the act and regulations pertaining to "organising and supporting the production and supply of housing"; (3) the act and regulations pertaining to "supporting the revitalisation, regeneration and renovation of dysfunctional and inefficient urban contexts"; and (4) the acts and regulations pertaining to the "rehabilitation, improvement and reconstruction of inefficient urban contexts" (KH-SS).

#### 5.1.1. Department for Roads and Urban Development

With the establishment of the 'national revitalisation working group' in 2013, the organisation has studied, designed and approved public policies in historic cities (N-SS). The department has allocated significant funds for pilot projects to regenerate life and raise cultural awareness inside traditional contexts (KH-SS). In historic cities, the local revitalisation working group employs qualified consultants to conduct relevant studies. Commissioned consultants are required to provide reports by conducting fieldwork, interviews and so on to propose suitable master plans. The commissioned consultants are also required to collaborate with government agencies to reflect their organisational perspectives into the proposed policies, in conjunction with the broader strategic plans (H-SS). A positive outcome of the recent regulations in historic cities is the formation of new cultural and shopping complexes, combined with façade restorations inside historic fabrics (Figure 3). Such developments can be viewed as a catalyst, and are expected to activate neighbouring private projects (N-SS, T-HD). Currently, the department's public policies, in conjunction with those of other government agencies or the private sector, may include: (1) direct interventions by the department, (2) the provision of financial incentives (e.g., loans, tax rebates, discounts and exemptions), and (3) cooperation with other investors, either by providing land or funds (H-SS). Recently, the department has implemented various investment packages in cooperation with local trust bodies and the private sector to address the local needs of the neighbourhoods (KH-SS).

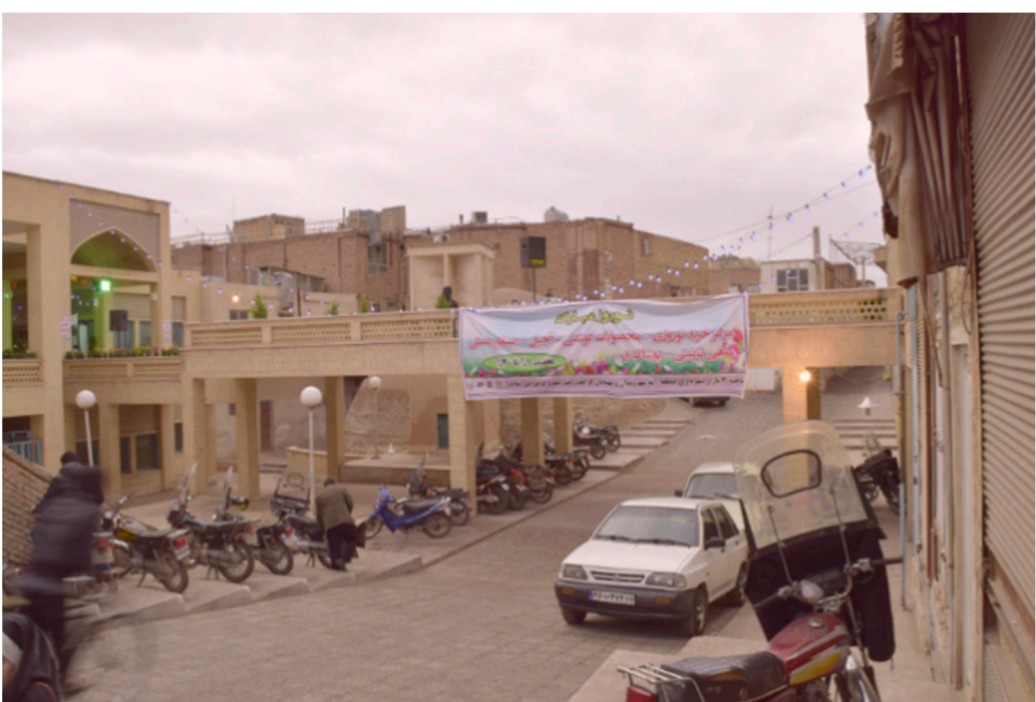

**Figure 3.** Development of DABs by implementing a shopping centre in historic Kashan, 2018 (Source: the authors).

### 5.1.2. Local Municipalities

To date, the municipality has rejuvenated several cultural–historic axes in the historic cities by cooperating with other agencies. In particular, as a means of utilising the DABs, some public projects have been proposed as new strategic plans (F-DM, GH-HUDD). The DABs are presently reflected in the sixth national strategic plan for the rehabilitation, improvement, reconstruction and empowerment of deteriorated−dysfunctional urban fabrics. This scheme requires the municipalities to follow a comprehensive five-year plan for the management of cross-agency policies (GH-HUDD). For instance, under the auspices of the Ministry of State, 270 neighbourhoods must be revitalised each year, in which some neighbourhoods should be located in historic neighbourhoods (H-HDRID).

The recent legislation initiated by the Ministry for Roads and Urban Development needs to be implemented by ICHHTO and municipal bodies. This new legislation targets the revitalisation of 2500 hectares of deteriorated fabrics nationwide, including DABs (F-DM). The methods for redeveloping DABs in municipalities are based on the following steps: (1) conducting feasibility studies to understand the urgent needs and setting a realistic definition of the projects (led by a hired qualified consultants); (2) selecting and clarifying the scope of the projects; (3) facilitating local offices to obtain public feedback; (4) providing master plans, as recommended by consultants based on the directions of the municipality or other agencies; and (5) the implementation of the proposed master plans by a municipality (F-DM).

At the municipality level, the aim is to generate pilot projects in neighbourhoods on a case-to-case basis as a means of attracting private investors and creating a cultural campaign for the public recognition of the historic areas. Investment packages also need to be defined, including the projected cash flow and income, profit, interest and so on. Additionally, basic infrastructures (e.g., water, electricity and so on) that are free of charge need to be provided by relevant government agencies. The local offices formed in the targeted neighbourhoods also need to connect private investors and property owners (F-DM). Along with other agencies, the municipality is expected to provide programmes to restore the socio-cultural axes and re-establish the neighbourhoods (Figure 4), consequently connecting several important historic sites that may branch out to other adjacent zones. A general assumption

is that such a linear approach to the projects can stimulate further development that mainly includes paving upgrades, façade preservation and the restoration/adaptive reuse of historic buildings, thereby injecting activities into the historic fabrics (S-SS).

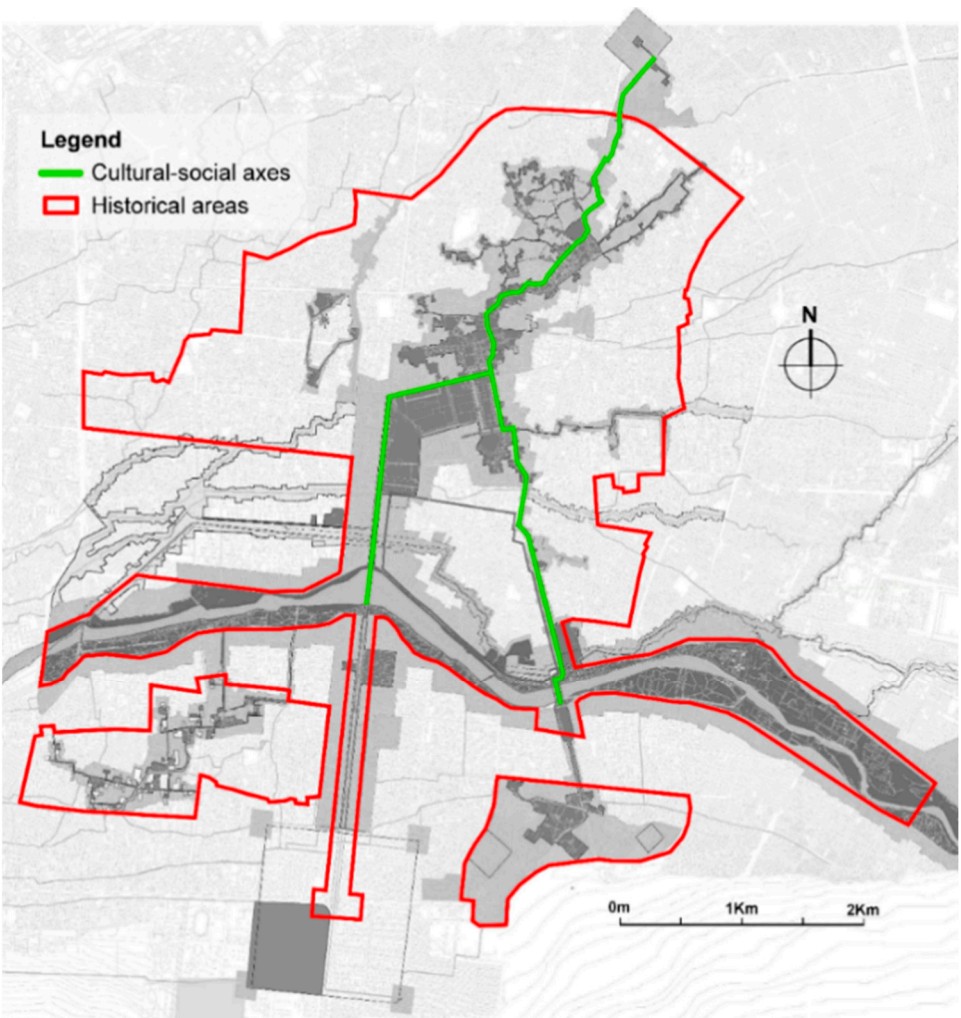

**Figure 4.** Proposition of the cultural–historic axes of Isfahan (map developed based on the strategic plan of Isfahan by NJP Consultants, 2006 revision [50]).

### 5.1.3. Iranian Cultural Heritage, Handicrafts and Tourism Organization

The heritage authority is expected to hire qualified consultants to study, design and evaluate essential projects in DABs. The Ministry for Roads and Urban Development acts as a supervisory agent to facilitate funding and cooperation for large-scale projects (M-UP, KH-HPRD). Municipalities are also expected to work closely with the ICHHTO and facilitate several large-scale mutual projects (B-UP, N-HRC), such as the rejuvenation of cultural–historic axes in Yazd, Kashan and Isfahan (Figure 4), as well as small-scale programmes, such as the restoration of neighbourhood centres (M-UP, KH-HPRD). The present role of the ICHHTO, along with the other two government agencies, can be specified as (1) introducing cultural–historic axes and (2) facilitating tourism investment packages for property owners and relevant stockholders (Z-HPRD).

### 5.2. City Fringe Developments versus Historic Urban Areas

In the modern era, financial and economic tradeoffs at the national level have principally caused historic fabrics to lose their strategic dominance in many Iranian cities [33]. The corresponding financial return of building investments is much more convenient when urban developments are located outside old cities. Therefore, city fringe developments can

guarantee ease of construction and ensure greater building density, lower building costs and reliable vehicular accessibility, which can significantly meet market demands. Thus, today's building investment is largely directed towards the city fringe, whilst central areas in historic fabrics remain underdeveloped (B-PUDP, B-UP, N-HRC, H-SS). In recent years, improper strategic planning has unrealistically broadened city boundaries and indirectly devaluated the lands within the central areas of cities. For example, a view of the map of metropolitan Isfahan (Figure 5) indicates that the city has experienced unprecedented urban sprawl in recent years (M-UP, KH-HPRD).

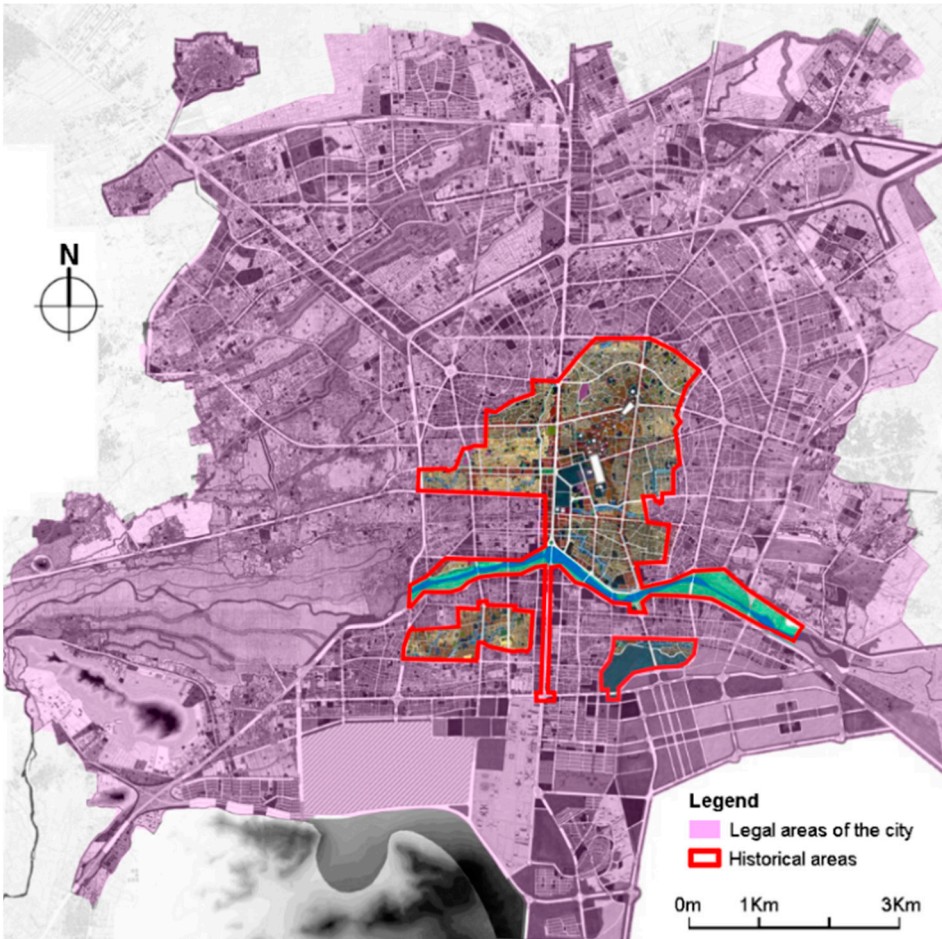

**Figure 5.** Strategic plan of Isfahan proposed in 1996. The historic core is demarcated by a solid red line in the centre (map based on the strategic plan of Isfahan by NJP Consultants [50]).

### 5.2.1. Inefficiency of Current Public Policies

Currently, the development plans are not updated because of their long processing time. For instance, the current development plan of Kashan took more than 16 years to be approved, and the implementation is not even expected for the next 18 years (N-SS, T-HD, Z-HPRD); moreover, during the same period, stakeholders rarely had the opportunity to offer their feedback (GH-HUDD, H-HDRID, KH-HPRD, M-UP). A significant problem in Isfahan and Kashan is related to the lack of an exclusive strategic plan for historic urban areas. As a result, historic areas may be ignored, neglected, or severely damaged (H-SS). In the proposal of strategic plans, environmental and financial aspects should match each other. Unfortunately, the financial aspects marginalise the environmental design qualities, which can destroy historic urban fabrics and even leave many of the selected/preserved historic sites freestanding, with no connection to their surroundings (B-PUD, GH-HUDD, M-PTA). In conclusion, urban management systems are unable to transfer such excessive

urban sprawl to DABs, which could have partially controlled the outward growth of cities (Figure 6).

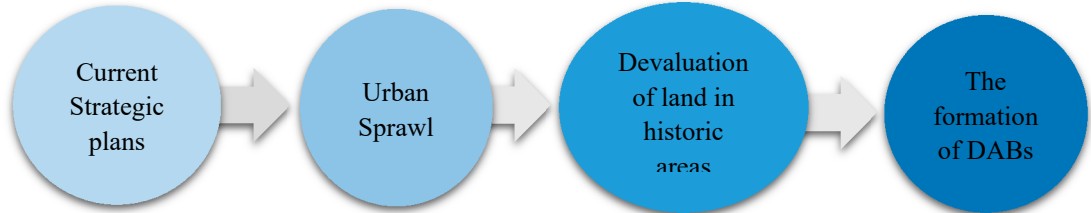

**Figure 6.** The process of generating DABs as a result of the current strategic plans in historic Iranian cities.

### 5.2.2. Governance Ambiguity and the Formation of DABs

The current shortage of urban incentives has discouraged residents from building/restoring their properties (Z-HPRD). No effective method has been planned to facilitate the investment, rejuvenation and provision of necessary infrastructures in the historic fabrics (B-UP, N-HRC). Meanwhile, dysfunctional acts, laws and legal obligations, private property ownership and inheritance regulations have contributed to the materialisation of DABs (GH-HUDD, H-HDRID, H-SS). In this aspect, government agencies should be accountable, as they cannot meet the contemporary needs of residents, and this scenario has arguably created a variety of urban, social and financial problems (KH-HPRD, M-UP, S-SS).

### 5.2.3. The Lack of Public Services and the Morphology of Historic Cities

Currently, the constant change in public attitudes and the prevalence of modern lifestyle have forced people to leave the historic areas. A historic house is widely known today as one that cannot satisfy contemporary human needs, and the maintenance of traditional houses can be time-consuming and expensive (Z-HPRD). Along with the scarcity of public services and civic infrastructure (e.g., medical centres, open spaces and so forth), the lack of vehicular accessibility has devaluated the lands (B-PUD), which can be considered a key reason for the lack of a sense of belonging to a place, as well as the excessive depopulation of historic areas (KH-HPRD, M-UP, Z-HPRD).

### 5.2.4. The Emigration of the Original Residents and the Immigration of Low-Income Communities

Currently, government agencies cannot facilitate the necessities of modern life for historic houses, a scenario that has pushed local owners to sell or abandon their homes (H-HDRID), which can eventually attract foreign refugees or disadvantaged communities to move into the devaluated and deteriorated fabrics (F-DM, KH-SS, S-SS). Moreover, the disadvantaged and low-income communities who reside in deteriorated historic fabrics cannot afford to restore their dilapidated homes, and they start to subdivide traditional properties, further attracting larger low-income populations. Furthermore, government agencies cannot sustain the provision of public infrastructure to new residents (N-SS, T-HD). Therefore, today's low-income immigrants are considered a threat to the preservation of historic areas (GH-HUDD, H-HDRID).

### 5.2.5. The Lack of Cultural Awareness and the Formation of DABs

In recent decades, the formation of new developments within the city fringe has generated negative cultural–psychological perceptions about historic cities amongst the public (B-UP, N-HRC). Many residents/owners have apparently demolished their traditional homes (e.g., by discharging water into the building foundations), with the hope of avoiding the heritage authority and its restrictive regulations, in order to be able to freely sell their properties (M-UP). The buyers of such properties are mostly interested in building multistorey apartments that can ruin the desired character of historic areas. In other words,

present-day owners remain unaware of the tremendous cultural–financial benefits that may be generated from the preservation of their old houses (M-UP, KH-HPRD). Such socio-cultural stigmatisation and public prejudice regarding historic areas can be viewed as worse than financial devaluation; as a result, Iranian layman historic areas may come to be treated as equal to problematically inferior districts (B-PUD, D-GC, M-PTA).

### 5.2.6. Generational Change and the Redevelopment of DABs

After a few hundred years, a single building in a historic area can be inherited by a large number of heirs, which contributes to ownership problems. The number of inheritors can accumulate over time, leading to property divisions, disagreement, abandonment and the generation of further DABs (M-UP, KH-HPRD). For example, the death of individual shareholders and ambiguous certificates or titles can cause further ownership complications in the redevelopment of DABs (GH-HUDD, H-HDRID, Z-HPRD). Studies on historic Kashan and Yazd have shown that the emigration of the original residents from historic areas (at least after two generations) has become inevitable (B-PUD). Thus, property owners are mainly individual inheritors who currently live outside historic areas and have a feeble sense of place–identity regarding their traditional homes (Figure 7) (B-UP, M-PTA, N-HRC). Moreover, in many circumstances, government agencies do not know the owners of DABs. Therefore, private ownership within the historic areas can delay the reconstruction of DABs (GH-HUDD, H-HDRID, KH-SS).

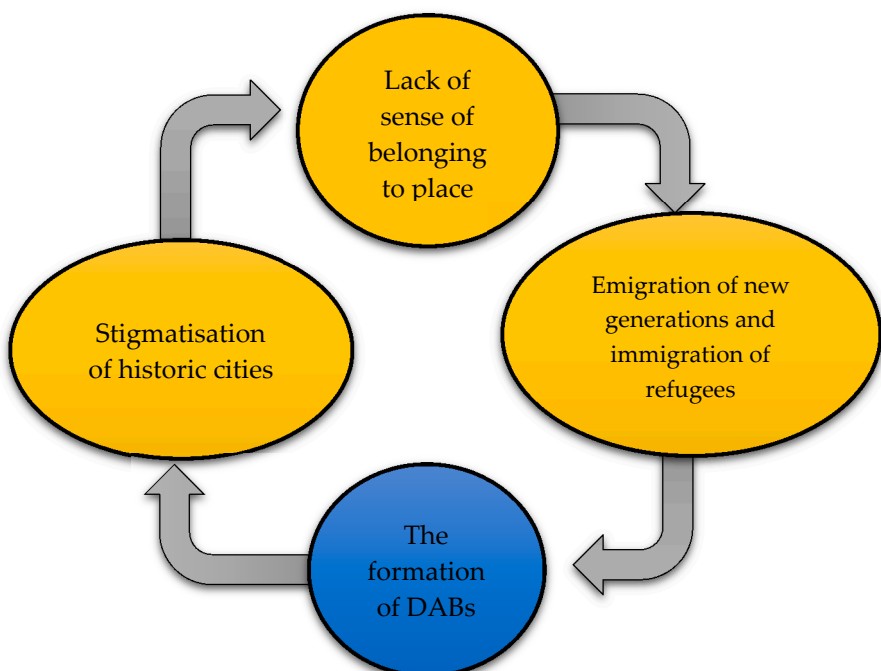

**Figure 7.** A faulty circular phenomenon may occur as a result of spatial–cultural problems, and can generate further DABs in historic cities.

### 5.3. Negative Aspects of Contemporary Practices

In the current planning context, many large-scale projects have been detrimental to the historic fabrics. The Imam-Ali project in Isfahan is a prominent example. Instead of redirecting vehicular accessibility to bypass historic areas, the project has allowed vehicles to traverse the historic context and thus jeopardise it. Today, the structural integrity of Masjid-i-Jame is threatened by the adjacent vehicular traffic (Figure 8). This condition can encourage the further use of cars, whilst many retailers who previously rode their bicycles use their private cars today to commute to work (S-SS). Such approaches also foster a negative public expectation regarding further road widening and car accessibility, and they have unequivocally harmed historic fabrics (KH-HPRD, M-UP, S-SS).

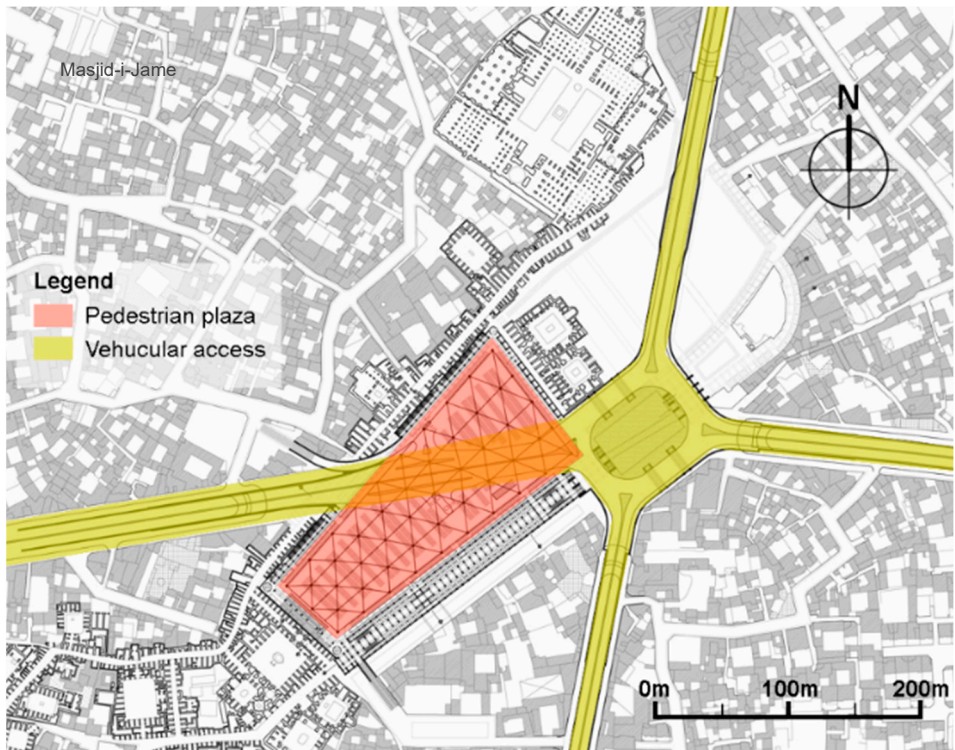

**Figure 8.** The Imam-Ali project is a large scale renewal program implemented in historic Isfahan (map developed based on the proposed plans by NJP Consultants, [50]).

### 5.3.1. Different Organisational Perspectives

In the current socio-spatial planning context, the mutual relationship amongst the three government stakeholders is not adequately defined. This lack of synchronisation has damaged historic fabrics, discouraged public life and created chaos in the spatial management of cities. Inappropriate building materials, unfitting building heights, unsuitable demolition of historic fabrics and unnecessary road widening have depopulated the historic zones (Z-HPRD). For instance, the Department for Roads and Urban Development has identified five types of problematic urban contexts, namely, 'historic fabrics, informal settlements, in-between contexts, rural backgrounds and former military camps' [41]. Nonetheless, DABs within historic cities can be categorised as both historic and in-between contexts, which in many circumstances may remain unspecified (N-SS, T-HD). This lack of cross-agency consistency has caused a low level of trust amongst private stakeholders (H-SS, H-TAD, M-PTA, B-PUD), whilst efficient restoration projects are rarely funded by financial institutions (M-UP, KH-HPRD).

### 5.3.2. Lack of Assessment Tools

Amongst the government agencies, an effective, sensible procedure at work is currently lacking for the assessment of the socio-spatial impact of urban regeneration programmes/policies (KH-S). The current public policies are mainly based on governmental perspectives, whilst the evaluations are falsely based on organisational propaganda instead of an accurate assessment of real outcomes. Today, one can identify many redevelopment projects that have won national bidding and are considered 'successful' but have not passed a realistic evaluation test (Z-HPRD). Many of the recent government projects that are claimed to be successful have never been accurately assessed, and their so-called achievements are seriously doubted. Thus, realistic assessment tools should be devised to measure the real impact of urban public policies (S-SS).

### 5.3.3. The Plans Are Physical and Do Not Infiltrate Traditional Fabrics

The government agencies have selected to restore several cultural–historic axes to revitalise historic urban areas. The policies mostly focus on façade restoration and the improvement of the stone pavements in predefined thoroughfares (Figure 9), which are either partially completed or superficial (H-SS). Currently, instead of proposing a combination of socio-spatial regulations, the mere 'physical policies' have substantially remained linear and freestanding, and therefore cannot generate life in the city (Z-HPRD). Meanwhile, beyond the restored frontages and pavements, the problem of DABs remains unresolved (KH-SS). The current urban public policies cannot be considered effective if they do not consider the social and financial grassroots of the historic neighbourhoods (M-PTA, B-PUD).

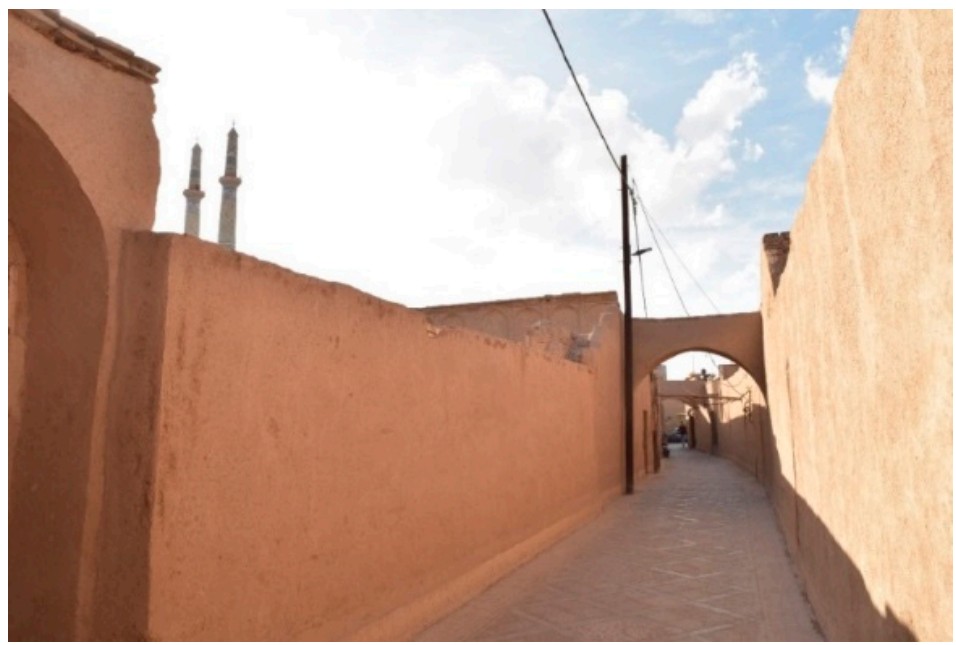

**Figure 9.** Restoring cultural and historic axes in Yazd (source: the authors).

### 5.3.4. The Agendas Are Non-Holistic and Are Not Incorporated into the Broader Context

The government agencies have not generated holistic plans whilst facilitating independent developments/renovations in historic areas. Such programmes/incentives also suffer from being implemented in a political atmosphere without having a full understanding of the larger built environments (M-UP, KH-HPRD), which neither interconnect neighbouring projects nor are incorporated within the surrounding urban contexts (F-DM). Such freestanding projects aim to attract investors by facilitating the adaptive reuse of segregated historic buildings for coffee shops, restaurants, hotels and so on. However, this approach cannot succeed without the incorporation of the surrounding broader context. Thus, colonnades, narrow alleyways, squares and adjacent fabrics need to be restored and incorporated into the proposed projects (S-SS).

### 5.3.5. The Plans Are Not Comprehensively Studied, and Are Mostly Implemented by Incompetent Consultants

Another problematic aspect of public initiatives/policies is the lack of adequate socio-spatial studies to implement the urban projects. In many cases, the regeneration programmes implemented by government agencies are mainly unsuccessful in terms of ensuring heritage preservation measures (i.e., destructing and road widening), whilst the impact of such adaptive reuse (e.g., tourist accommodation) is largely overlooked (M-UP). Moreover, in several environments, an urban activity overload generated by new developments in historic areas has negatively affected historic fabrics. For instance, high levels of vehicular and human traffic can impose irreparable damage to heritage sites

in Kashan (Z-HPRD). Thus, despite producing a large number of strategic and master plans, historic urban fabrics have never been realistically understood, enhanced or restored (M-UP, KH-HPRD, F-DM). As a result, we need to provide realistic socio-spatial studies and define several policy areas specific to each historic city. The proposed plans should not generalise problems, as many historic cities have traditional bazaars that have helped to preserve their financial centrality (e.g., Kashan, Isfahan); meanwhile, in other cities (e.g., Shiraz), the bazaar has lost its significance (M-PTA, B-PUD). The urban policies and procedures that have been produced as a result of specific studies must support the broader socioeconomic context (D-GC, KH-SS).

### 5.3.6. The Lack of Social Capacity Building

Government agencies seem to only partially understand the context of historic cities, and they implement projects/programmes without proper community engagement. Subsequently, effective projects and policies cannot be implemented. The systematic organisational problems have clearly prevented community participation and have weakened public policies [51,52]. This lack of social capacity building can be mainly attributed to the top-down planning and decision-making approach that is implemented at the national level by government agencies (H-SS, KH-HPRD, M-UP). The lack of public participation has also reduced land prices and affected local investments/development opportunities in historic areas (M-UP, KH-HPRD, H-SS). Hence, the relevant agencies need an extensive plan to heed the real sentiments of the inhabitants and generate local empowerment. Moreover, government stakeholders need to implement community participation programmes before and during the implementation of public projects, as they can reflect the real needs of the targeted neighbourhoods (M-UP, KH-HPRD).

### 5.3.7. The Displacement of Residents by Urban Public Projects

In many public initiatives, government agencies have essentially displaced the original residents. For instance, in the context of the Jammalleh project in Isfahan, the original residents were forced to sell their properties and evacuate the neighbourhood. After the completion of the project, the newly built units/shops remained unoccupied for a long time and eventually became abandoned buildings (M-UP, KH-HPRD). Another example is the Imam-Ali megaproject that was implemented by the Isfahan municipality (Figures 8 and 10). The project began in 2007, and the municipality has already spent a considerable amount of public funds, but the second phase has yet to be started. The general policy focused on the possession of all historic land and properties within the boundaries of the regeneration project (S-SS). Consequently, even if the original shop owners were forced to sell their properties, the municipality did not gain sufficient demand to sell or rent out the newly built commercial areas, resulting in the large number of closed shops despite the government's massive expenditure (S-SS). As a result, the public sector currently suffers from a lack of financial resources (i.e., investment) because many projects cannot meet the anticipated economic requirements (S-SS).

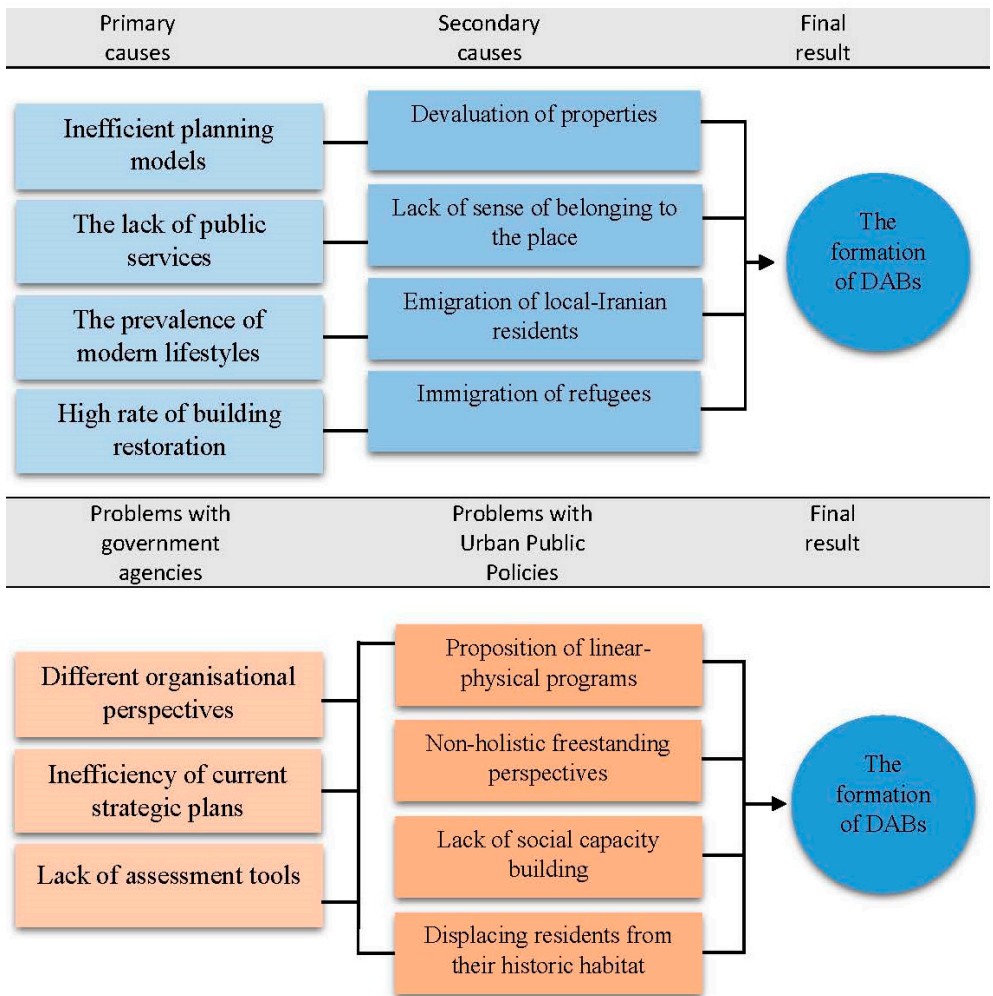

**Figure 10.** DABs and urban public policies in historic Iranian cities (based on Sections 5.2 and 5.3).

## 6. Discussion

This study elaborated the results of the in-depth interviews on two levels. In the first part, the socio-spatial causes that have generated DABs and characterise their impact on historic fabrics were identified. The second part demonstrated the current public policies and practices that have yet to be implemented by the three government agencies. Several shortcomings can be specified from the critical analysis. By singling out the extreme proliferation of DABs in the historic cities, this study illustrated the unsuccessful aftermath of urban public policies in Iran and in a wide variety of cases in historic cities. The research subsequently offered innovative contributions in a number of areas. In historic cities, the emigration of original residents is generally perceived as the focal reason for the formation of DABs and the devaluation of land. Within the context of urban planning, it was deliberated how the current extent of DABs can be related to the larger design–planning context that encourages urban sprawl. The interviews showed how current public policies and initiatives are largely engaged with the visual–physical aspects of cities. Seeking to verify this claim, the interviews were able to elaborate upon how the policies and initiatives have been non-holistic, linear, and physical in most circumstances; they are also mostly concentrated on the regeneration of freestanding urban axes/nodes, and have utilised façade and paving restorations rather than considering the grassroots of social life [53].

This research is the first step towards a more profound understanding of the socio-spatial equations in Middle Eastern and Iranian historic cities. The findings will allow practitioners, academics and policymakers to investigate the current problems in real-life

contexts. This research has further clarified how the lack of effective assessment tools, combined with contradictory organisational perspectives, has generated urban policies that underestimate social capacity building and displace the original residents from their natural habitats. This research has also elaborated upon how the government programmes have been chiefly designed as freestanding entities which are incapable of being interconnected with their broader urban context. The enormous growth of DABs reconfirms that, despite the disbursement of huge amounts of public funds by government agencies, the current socio-spatial planning context in historic Iranian cities has largely remained unsuccessful (Figure 10, bottom).

As discussed in Section 1.1, the actual geographical, social and political contexts of historic cities in Middle Eastern and European–Western cities generate enormously different built environs. This scenario has completely changed the way those nations have protracted their inner-city transformation projects, policies and processes. Such spatial-historic differences can be viewed as the central reason why Western models of urban regeneration cannot yield an acceptable level of outcome in Iranian or Middle Eastern historic cities [7]. By comparing the experiences of the Middle Eastern nations and the Western world, one can identify a gap in knowledge, in which the argument, approach and methodology introduced by Western schools of thought must be enhanced and fine-tuned before they can be applied and efficiently address the problem of DABs in historic cities in Iran and the Middle East [10]. Consequently, this section investigates the feasibility of several types of factors to control and/or minimise the impacts of current public policies on the formation of DABs. On this basis, the study scrutinises the relevant urban policies in several Middle Eastern historic cities. Such real life experiences can provide valuable lessons to be learnt for controlling DABs in historic Iranian cities and/or within reasonably similar contexts in other Middle Eastern countries.

### 6.1. Policy and Regulations

By looking into the experiences of other historic cities, several lessons can be learnt with regards to the provision of effective policies and regulations, which were already stated throughout the interviews and data analysis in this study. For instance, the rehabilitation plan of DABs in old Rusafa, Iraq, as submitted by Japanese consultants (JCP) in 1984, did not consider citizen participation as the central element in making decisions [54]. Investigated within the Asian context, the research indicated that contemporary problems in the historic area of George Town, Malaysia encompass the depopulation of the inner-city, and they have caused intensive and uncontrolled development pressures, changing lifestyles and irregular consumption patterns of city dwellers, insufficient legislation, and environmental degradation, which altogether would threaten the existence of the historic city [55]. Therefore, public policies must be proposed/assessed by competent consultants to facilitate the generation of social capital amongst all stakeholders, particularly the original residents and property owners, to hinder the displacement of residents from their historic habitat. Government agencies must ensure that a fair deal is achieved by both the local owners and possible external investors. This scheme is in agreement with the recommendations on the refurbishment of dilapidated heritage buildings in George Town, Malaysia [56].

### 6.2. Implementation and Tools

The interviewees believed that the lack of vehicular accessibility is a major factor encouraging the emigration of the original residents and the concomitant antisocial behaviours inside the historic city, hypotheses which are both supported by the literature [18,55]. By referring to Middle Eastern experiences, the rehabilitation of Hasifa, Tunis and the real-life experiences both demonstrate the importance of facilitating vehicular accessibility and local infrastructure, which can be considered the most crucial task of preventing the spread of DABs in historic areas. Exclusive traffic master plans need to be devised for historic neighbourhoods, especially when new developments have to be

applied within DABs. New legal tools also need to be considered and approved to be able to resolve and simplify ownership problems. For instance, there is a strong need to facilitate regulations that can assure a fair trade between investors and unknown or multiple owners/shareholders in historic cities [57]. Consequently, the relevant best-practice can regenerate historic areas by providing suitable street networks, intersections, parking lots, urban infrastructures and custom-made public transportation facilities [58].

### 6.3. Administration and Supervisions

As has been learnt from the past experiences of the rehabilitation of historic cities [59,60], there is an urgent need to propose integrated management tools and incentives (e.g., specific detailed master plans for historic areas, aligned with informed strategic plans) that are equally legible and applicable at all levels of government agencies to satisfy the different organisational perspectives whilst engaging stakeholders to halt the devaluation of land and the emigration of residents. The agencies need to realistically reassess the public policies and document/record the historic fabrics.

### 6.4. Financial Resources

The results indicate that government agencies need to set up local offices before and during the implementation of public initiatives and policies within the targeted neighbourhoods in view of the facilitation of real participation [61], as this approach can generate capacity building amongst all stakeholders and reflect the real needs of the inhabitants. The agencies should cooperate with a variety of public–private investors, including original (business) owners, to create jobs, generate business and inject life into historic cities.

As for future strategic plans, on the basis of the literature [62], DABs can be perceived as a wider opportunity to meet the needs of the original residents, and to accommodate new urban developments (i.e., public services, open space or new vehicular roads, car-parks and so forth) to regenerate the peripheral urban areas. There is a great need to provide a reliable data bank that can facilitate realistic investment packages and attract public and private investment.

## 7. Conclusions

The interest in dealing with issues concerning DABs towards achieving resilient historic locales at the policy-making levels is a relatively recent one [63]. This study has verified the devaluation of properties in historic cities due to the current inefficient urban public policies combined with the lack of vehicular accessibility, the prevalence of modern lifestyles, the lack of cultural awareness and the high rate of building restoration. This research argues that such a devaluation of land can generate a lack of a sense of belonging amongst residents, thus causing them to leave the historic zones [37]. The emigration of the original residents can further trigger secondary consequences and may attract low-income disadvantaged communities to settle in the abandoned areas as a means of accessing cheaper housing opportunities. An examination of the socio-spatial layers can help to unfold the substantial correlations between urban public policies and the formation of DABs in historic cities (Figure 10, top).

The participants' perceptions, as reflected in this study, reiterate the solutions that can be deemed valuable for planners and decision-makers in both developing and developed nations where old cities generally face intensified urban problems as a result of rapid economic development and population growth. The results and discussions in the research are in line with the idea that the substantial economic growth and modernisation of many emerging countries necessitate the reassessment of governing views with respect to the impact of urban policies and planning tools as a means of reducing inequality and enhancing local development [64]. The modification of urban public policies in historic cities can reduce social segregation, as the task involves incorporating the views of both the original residents and government stakeholders in the governing process [65]. The discussions reconfirmed the theoretical and practical inadequacies of the current public

policies in historic cities, which have failed to consider the social, cultural, physical, political and economic aspects or the stakeholders' aspirations, which has weakened the sustainable continuity of the local culture in the historic urban environment [66].

The analysis of the results culminated with a set of recommendations pertaining to new types of well-studied regulations that can holistically address DABs and facilitate social capital amongst all stakeholders. Furthermore, morphologically informed initiatives should be urgently developed within DABs to holistically penetrate the forgotten heritage fabrics and improve the lost sense of belonging to a place in the historic cities. This aspect requires strict policies to control the urban sprawl. Moreover, the proper facilitation of vehicular accessibility and local infrastructure is regarded as the most crucial task for halting the spread of DABs in historic areas, as it can strengthen the sense of belonging to a place [67]. Consequently, the policies discussed in this work should emphasise the need to attain reasonable accessibility and prevent an overload of vehicular movement throughout the subjected areas. This research is meant to help decision-makers create efficient urban public policies in the historic cities of Iran. In particular, the proposed measures aim to reduce the number of DABs, and they can further help other Middle Eastern government agencies to propose integrated management tools for the stimulation of historic cities.

However, several inevitabilities demonstrate the need to re-examine the current discussion. One of the limitations of this work is the number of case studies and participants that were investigated. Further interviews need to be conducted amongst a larger number of qualified participants in other historic cities in order to be able to disclose the other unknown cultural, social or financial impacts of urban public policies with respect to the formation of DABs. Although the research has recounted a substantial argument for redeveloping DABs in historic cities, other cultural and financial factors also need to be investigated and regulated. The questions could fit into the following themes: In whose interest is it for new housings/buildings to be developed within DABs? Why should public–private developers be interested in new projects unless current undesirable populations are removed? Do DABs represent an opportunity to implement the required infrastructure or a new form of affordable housing? Studies on the cultural aspects of historic cities may only be detrimental to future researchers who pursue the implementation of in-between spaces as a morphologically informed method to revitalise DABs in historic urban areas.

**Author Contributions:** Conceptualization, H.T.; methodology, H.T., M.H.M.; validation, H.T.; formal analysis, H.T., M.H.M.; investigation, H.T.; resources, H.T., M.H.M.; data curation, H.T.; writing—original draft preparation, H.T., M.H.M.; writing—review and editing, H.T., M.H.M.; visualization, H.T.; funding acquisition, M.H.M. All authors have read and agreed to the published version of the manuscript.

**Funding:** This research was funded by the Universiti Sains Malaysia (grant number RUI 1001/PP-BGN/8016079).

**Institutional Review Board Statement:** The study was conducted according to the guidelines of the University of Adelaide, and was approved by the Office of Research Ethics, Compliance and Integrity, The University of Adelaide (protocol code H-2018-047 on 9 March 2018).

**Informed Consent Statement:** Informed consent was obtained from all of the subjects involved in the study.

**Data Availability Statement:** The data presented in this study are available on request from the corresponding author. The data are not publicly available due to restrictions e.g. their containing information that could compromise the privacy of research participants.

**Acknowledgments:** The authors would like to thank Universiti Sains Malaysia for providing financial supports under an RUI grant (RUI 1001/PPBGN/8016079).

**Conflicts of Interest:** The authors declare no conflict of interest.

# Appendix A

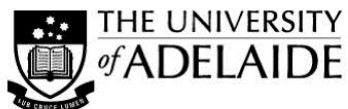

Our reference 32794

09 March 2018

Associate Professor Julian Worrall
School of Architecture & Built Environment

Dear Associate Professor Worrall

**ETHICS APPROVAL No:**  H-2018-047
**PROJECT TITLE:**  Application of spatial liminality in urban design, towards an approach for revitalising unexploited land areas historical Iranian cities

The ethics application for the above project has been reviewed by the Low Risk Human Research Ethics Review Group (Faculty of Arts and Faculty of the Professions) and is deemed to meet the requirements of the *National Statement on Ethical Conduct in Human Research (2007)* involving no more than low risk for research participants.

You are authorised to commence your research on:     09/03/2018
The ethics expiry date for this project is:     31/03/2021

**NAMED INVESTIGATORS:**

Chief Investigator:     Associate Professor Julian Worrall

Student - Postgraduate     Mr Hamed Tavakoli
Doctorate by Research (PhD):

Associate Investigator:     Mr Ehsan Sharifi

Associate Investigator:     Dr Nigel Westbrook

**CONDITIONS OF APPROVAL:** The revised application provided 06.03.2018 and the revised table 1 and table 7 provided 08.03.2018 have been approved.

Ethics approval is granted for three years and is subject to satisfactory annual reporting. The form titled Annual Report on Project Status is to be used when reporting annual progress and project completion and can be downloaded at http://www.adelaide.edu.au/research-services/oreci/human/reporting/. Prior to expiry, ethics approval may be extended for a further period.

Participants in the study are to be given a copy of the information sheet and the signed consent form to retain. It is also a condition of approval that you immediately report anything which might warrant review of ethical approval including:

- serious or unexpected adverse effects on participants,
- previously unforeseen events which might affect continued ethical acceptability of the project,
- proposed changes to the protocol or project investigators; and
- the project is discontinued before the expected date of completion.

Yours sincerely,

Dr Anna Olijnyk

Convenor

Dr Jungho Suh
Convenor

The University of Adelaide

**Figure A1.** Ethics approval.

## Appendix B

**Table A1.** List of interviewees and the references.

| Interviewee Number | Interviewees Relevant Organizations | Organizational Responsibility | Names and Affiliations as Are Referenced in This Article |
|---|---|---|---|
| 1 | Iran Cultural Heritage Handicrafts and Tourism Organization (ICHHTO, Kashan) | Head of Preservation and Regeneration Department | Z-HPRD |
| 2 | Iran Cultural Heritage Handicrafts and Tourism Organization (ICHHTO, Yazd) | Urban Planner | B-UP |
| 3 | Iran Cultural Heritage Handicrafts and Tourism Organization (Yazd World Heritage Centre, ICHHTO, Yazd) | Head of Research Centre | N-HRC |
| 4 | Iran Cultural Heritage Handicrafts and Tourism Organization (ICHHTO, Isfahan) | Head of Preservation and Regeneration Department | KH-HPRD |
| 5 | Iran Cultural Heritage Handicrafts and Tourism Organization (ICHHTO, Isfahan) | Urban Planner | M-UP |
| 6 | Ministry for Roads and Urban Developments (Kashan) | Senior Staff | N-SS |
| 7 | Ministry for Roads and Urban developments (Kashan) | Head of Department | T-HD |
| 8 | Ministry for Roads and Urban developments (Yazd) | Senior Staff | KH-SS |
| 9 | Ministry for Roads and Urban Developments (Isfahan) | Senior Staff | H-SS |
| 10 | Yazd City Council | Deputy mayor | F-DM |
| 11 | Kashan City Council | Head of the Urban Design Department | GH-HUDD |
| 12 | Development, Renewal and Improvement Department (Kashan City Council) | Head of Development, Renewal and Improvement Department | H-HDRID |
| 13 | Development, Renewal and Improvement Department (Isfahan City Council) | Senior Staff | S-SS |
| 14 | Helli Private Design Studio (Kashan) | Traditional Architect and Developer, specialist on redeveloping DABs | H-TAD |
| 15 | Dastgah-saz Builders and Contractors (Yazd) | Head of the Company (General Contractor): specialist in redeveloping DABs in historic areas | D-GC |
| 16 | Isfahan Housing Development Corporation (Sherkate-i-toseh-i-mskan-i-Isfahan) | Head of the Company; Engineer and Developer | H-ED |
| 17 | Isfahan Housing Development (Sherkate-i-toseh-i-mskan-i-Isfahan)Corporation (Sherkate-i-toseh-i-mskan-i-Isfahan) | Senior Engineer and Developer | H-ED |
| 18 | Iran University of Science and Technology | Head of the School of Architecture; Principal Urban Designer (Theoretician in Historic Cities) | B-PUD |
| 19 | Iran University of Science and Technology | Professor (theoretician) in Islamic Traditional Architecture | M-PTA |

## Appendix C

**Table A2.** Interview Questions.

| **A. Interview with Policy Makers and Representative from the Three Government Agencies** |
|---|
| An invitation letter introduced the lead researcher to government agencies for conducting interviews. Upon the formal correspondences, several representatives in three cities were contacted and interviewed: |
| **Question 1:** Do DABs become a serious problem in historic cities of Iran? Please explain if your answer is yes. |
| **Question 2:** Why and how are DABs forming inside historic cities of Iran? |
| **Question 3:** Regarding large areas of abandoned and dilapidated properties inside historic urban fabrics, which programs/policies/design moves are directly addressing the underutilization of land inside historic cities? What are their cons and pros? Are they proved to be successful or not? How does your organization assess the historical ramifications of such programs and policies? |
| **Question 4:** How does you organization identify feasibility studies, and implement/audit public policies (e.g. for reutilizing DABs) in historic areas? |
| **Question 5:** Does your organization has any programs for reutilizing DABs in the future? How do such urban policy/design moves could attract or deter public or private building investments inside DABs? |
| **Question 6:** Do you have suggestions for revising current preventative policies-regulations? |
| **B. Interviews with practitioners (architects, urban designers, urban planners, contractors and developers)** |
| **Question 1:** Do DABs become a serious problem in historic cities of Iran? Please explain if your answer is yes. |
| **Question 2:** Why and how are DABs forming inside historic cities of Iran? |
| **Question 3:** Regarding large areas of abandoned and dilapidated historic urban fabrics, which programs and policies (by private of public institutions) are directly causing such massive deleterious phenomenon inside historic cities? For instance how does the lack of vehicular accessibility can play a role in the further dilapidation of historic areas? |
| **Question 4:** What are the best method of urban design/policy (e.g. simulative incentives) which can best attract sustainable building investments towards DABs and unexploited historic areas? What are the factors that can limit such building investments? |

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
