# Peer review of "Urban Public Policy and the Formation of Dilapidated Abandoned Buildings in Historic Cities: Causes, Impacts and Recommendations"

_sustainability, doi:10.3390/su13116178_

Round 1

Reviewer 1 Report

The study takes into consideration the theme relating to current public policies amongst essential key players and stakeholders, for ameliorating the problem of dilapidated-abandoned buildings (DABs), which through the literature proved to be associated with socio-spatial disadvantage.

The theme, in addition to being extremely interesting, is also very current as it affects most of the historic centers of contemporary cities. The topic is well treated especially for the part that concerns the social and economic aspects, but perhaps it would be necessary to consider other equally important aspects that lead to the abandonment of historic centers. It is perhaps necessary to implement some bibliographic references especially in the introduction, but also in section 2 to support the various statements, and in the conclusions.

There are several aspects that require more attention (especially materials and methods)

The objective of the research and the research questions are clear.

The literature present is summarized well in the text of the article, but perhaps some integration is needed which should be referred to in a timely manner in the introduction and discussions, and in general throughout the text.

In general, the article is understandable but repetitive in some points (sections 2 and 5 above all), the description of the different steps requires attention (introduction, materials and methods, results, discussion, and conclusion)

In particular:

Introduction

The introduction is interesting, but perhaps it would be necessary to expand this section a little, framing the problem in a broader overview. In fact, there is talk of a problem common to many other cities, not only Iranian. It is a problem that is found in the historic centers of Spain, Italy, and France. How are the policies of other countries? And then this phenomenon is based only on socio-commercial dynamics, or of degradation, or also on other events? For example, could a problem in historic centers not also be affected by the perception of structural instability in buildings? Maybe following natural events?

  1. The Research Context

The Iranian context is rightly well described in this section. The statements and concepts are clearly reported and reference is made to adequate bibliographic references. For this reason I would show in section i (introduction) a broader overview that frames the problem in its entirety (obviously everything must be supported by evidence from the literature).

Also in sections 2.1 and 2.3 I would deepen the problems that cannot only be linked to the morphological conformation of the historic city or to commercial dynamics. Unless you find that in Iranian cities, the issues you face are comprehensive.

  1. Public policies in historic Iranian cities

This section is clear and straightforward and appropriately describes the “problem” related to urban public policies.

  1. Data and Methods

The section does not clearly elaborate on the scientific method used for the research. It is not clear, for example, the method used to construct the questionnaire. It is not explicitly explained how the data is processed. This section does not provide clear details on the scientific / analytical method used for research analysis and results based on scientific analysis. The section requires great attention.

  1. Results

Row 199: section 4.1 what does it refer to? I can't find this section in the methods.

Section 5.1 very discursive. Since the scientific method has not been clarified in the previous section, this section appears too long and dispersed. I recommend summarizing the essential concepts.

Section 5.2: Some considerations (also present in previous section 5.1) could be moved to discussions. In this way the results would be more fluid in the reading.

Attention, several concepts are repeated (in a discursive way) already treated in section 2 (for example many concepts of paragraphs 5.2.3 and 5.2.4)

In general, the results should be reviewed, summarized and organized also according to a revision of the method in scientific form. An information but the results of the questionnaire? Where are they summarized? In general I cannot find fluidity in reading the results, it would be necessary, perhaps, to reorganize the research material.

  1. Discussion and Recommendations

In support of the numerous statements, it would also be necessary to refer to the scientific literature of the wide international panorama.

The reading was however formative and extremely interesting. Reading the document may interest readers from different areas because the topic lends itself to being studied across the board involving different disciplines

If you have any questions, please, contact me through the Editor.

Author Response

Response to Reviewer No.1’s comments:

The study takes into consideration the theme relating to current public policies amongst essential key players and stakeholders, for ameliorating the problem of dilapidated-abandoned buildings (DABs), which through the literature proved to be associated with socio-spatial disadvantage.

The theme, in addition to being extremely interesting, is also very current as it affects most of the historic centers of contemporary cities. The topic is well treated especially for the part that concerns the social and economic aspects, but perhaps it would be necessary to consider other equally important aspects that lead to the abandonment of historic centers. It is perhaps necessary to implement some bibliographic references especially in the introduction, but also in section 2 to support the various statements, and in the conclusions.

There are several aspects that require more attention (especially materials and methods)

The objective of the research and the research questions are clear.

The literature present is summarized well in the text of the article, but perhaps some integration is needed which should be referred to in a timely manner in the introduction and discussions, and in general throughout the text.

In general, the article is understandable but repetitive in some points (sections 2 and 5 above all), the description of the different steps requires attention (introduction, materials and methods, results, discussion, and conclusion).

We are humbled by Reviewer#1’s generous comments on our work. We thank her/him for having the confidence with our work.

In particular:

Comment (1.1): Introduction

The introduction is interesting, but perhaps it would be necessary to expand this section a little, framing the problem in a broader overview. In fact, there is talk of a problem common to many other cities, not only Iranian. It is a problem that is found in the historic centers of Spain, Italy, and France. How are the policies of other countries? And then this phenomenon is based only on socio-commercial dynamics, or of degradation, or also on other events? For example, could a problem in historic centers not also be affected by the perception of structural instability in buildings? Maybe following natural events?

Response to Comment (1.1): In response to this comment, kindly note that we have added Section 1.1. Geographical context and differences in inner-city transformation in historic Middle-Eastern and European cities to the introduction section. Also, new resources were added to back up the discussion (please refer to Section 1.1.).

Comment (1.2): The Research Context

The Iranian context is rightly well described in this section. The statements and concepts are clearly reported and reference is made to adequate bibliographic references. For this reason I would show in section i (introduction) a broader overview that frames the problem in its entirety (obviously everything must be supported by evidence from the literature).

Also in sections 2.1 and 2.3 I would deepen the problems that cannot only be linked to the morphological conformation of the historic city or to commercial dynamics. Unless you find that in Iranian cities, the issues you face are comprehensive.

Response to Comment (1.2): In chapter 2, two new paragraphs including new resources were added to reflect the broader debate and literature on historic cities in the world (please refer to Section 2).

In agreement with the reviewer’s comment, the focus of these subsections is on the Iranian cities. To improve the clarity, we have added Section 2.1. DABs in Historic Iranian Cities to highlight the issues in the context of Iran (please refer to Section 2).

Comment (1.3): Public policies in historic Iranian cities

This section is clear and straightforward and appropriately describes the “problem” related to urban public policies.

Response to Comment (1.3): We thank the reviewer for this observation.   

Comment (1.4): Data and Methods

The section does not clearly elaborate on the scientific method used for the research. It is not clear, for example, the method used to construct the questionnaire. It is not explicitly explained how the data is processed. This section does not provide clear details on the scientific / analytical method used for research analysis and results based on scientific analysis. The section requires great attention.

Response to Comment (1.4): As mentioned in Section 4, in-depth interviews were conducted in order to address the main research question. However, the table of abbreviation was missed in the previous version of the paper and now it is added as Appendix B, at the end of the text, including the abbreviated name of the interviewees (to protect the confidentiality), their relevant institutions/organizations and the roles of the interviewees within that organization. However, as mentioned in Section 5, the subsections in the Results Section have been extracted from the conducted in-depth interviews as major themes, and then rearranged to form each subsection.

Moreover, we have added Section 4.1 Analytic Techniques to improve the clarity (please refer to Section 4.1 and Appendix B). 

Comment (1.5): Results

Comment (1.5.1): Row 199: section 4.1 what does it refer to? I can't find this section in the methods.

Response to Comment (1.5.1): As it is truly mentioned by Reviewer #1, Appendix A was added, indicating the letter of ethics approval (please refer to Appendix A).

Comment (1.5.2): Section 5.1 very discursive. Since the scientific method has not been clarified in the previous section, this section appears too long and dispersed. I recommend summarizing the essential concepts.

Response to Comment (1.5.2): In response to this comment, kindly note that the subsections in section 5 have been extracted from the conducted in-depth interviews as major themes, and then rearranged to form each subsection. However, as suggested by the reviewer, we have added Table 1 to summarise themes and subthemes as extracted from the analysis. Meanwhile, we have tried our best to summarise each subsection to improve clarity, as suggested by the reviewer.

Response to Comment (1.5.3): Section 5.2: Some considerations (also present in previous section 5.1) could be moved to discussions. In this way the results would be more fluid in the reading.

Attention, several concepts are repeated (in a discursive way) already treated in section 2 (for example many concepts of paragraphs 5.2.3 and 5.2.4)

Response to Comment (1.5.3): As suggested by Reviewer #1, some of the sentences of Sections 5.2.3 and 5.2.4 to the discussion section (please refer to Results and Discussion Sections).

Response to Comment (1.5.4): In general, the results should be reviewed, summarized and organized also according to a revision of the method in scientific form. An information but the results of the questionnaire? Where are they summarized? In general I cannot find fluidity in reading the results, it would be necessary, perhaps, to reorganize the research material.

 Response to Comment (1.5.4): As has been mentioned in Section 5, three significant themes were developed based on thematic analysis of in-depth interviews, including (A) government agencies and current public policies, (B) city fringe developments versus historic urban areas, and (C) negative aspects of contemporary practices, as shown in Table 1 and discussed in Section 5 (please refer to Section 5).

Comment (1.6): Discussion and Recommendations

In support of the numerous statements, it would also be necessary to refer to the scientific literature of the wide international panorama.

Response to Comment (1.6): In agreement with the reviewer, the discussion section is enhanced with a major modification by implanting international experiences. First, it was deliberated why urban transformation in European and in general Western cities can be fundamentally different from the Middle Eastern counterparts. Later, several successful case studies (in a relatively similar context, i.e. the Middle Eastern cities) were added and analysed, including Tunis, Iraq, Egypt, Morroco, and Uzbekistan. The content then was analysed/compared and lessons learnt were extracted and elaborated in each section without numbering (please refer to Section 6).

The reading was however formative and extremely interesting. Reading the document may interest readers from different areas because the topic lends itself to being studied across the board involving different disciplines

We thank the reviewer for this observation.   

Reviewer 2 Report

Dear Authors,

I read the article with great interest. I see great potential in it, I also appreciate the contribution you put into it.
However, while reading the manuscript, several points caught my attention. I believe that an improvement to the article will be needed. For the sake of clarity, I will point out my doubts.

1. Chapter 2, which I see as a Research review, should be written in the context of the broader debate on historic cities in the world. Sustainability is an international journal and it would be good to find a context for similar processes and phenomena in other historic cities of the world. As you well know, the subject literature is really rich.
2. Chapter 4 should follow the chapter: Methods
3. The Discussion chapter should be rooted more in international discourse. Not all of it, of course. However, this is missing now. It would be worth showing what your research has contributed and what has not contributed to broadening the general knowledge about historical cities.
I believe that the listing of recommendations in the Discussion chapter is incorrect. Of course, the remediation theses are good. Nevertheless, the Discussion should be more confronted with different opinions on specific actions; present their strengths and weaknesses. Better to choose fewer and discuss them.
4. The Conclusions chapter is too long. Some of the questions/sentences in this chapter are more relevant to the Discussion.

Author Response

Response to Reviewer No.2’s comments:

I read the article with great interest. I see great potential in it, I also appreciate the contribution you put into it.
However, while reading the manuscript, several points caught my attention. I believe that an improvement to the article will be needed. For the sake of clarity, I will point out my doubts.

Comment (2.1): Chapter 2, which I see as a Research review, should be written in the context of the broader debate on historic cities in the world. Sustainability is an international journal and it would be good to find a context for similar processes and phenomena in other historic cities of the world. As you well know, the subject literature is really rich.

Response to Comment (2.1): In chapter 2, two new paragraphs including new resources were added to reflect the broader debate and literature on historic cities in the world (please refer to Section 2).

Comment (2.2): Chapter 4 should follow the chapter: Methods

Response to Comment (2.2): Ok, we have revised it based on the reviewer’s comment.

Comment (2.3): The Discussion chapter should be rooted more in international discourse. Not all of it, of course. However, this is missing now. It would be worth showing what your research has contributed and what has not contributed to broadening the general knowledge about historical cities.

I believe that the listing of recommendations in the Discussion chapter is incorrect. Of course, the remediation theses are good. Nevertheless, the Discussion should be more confronted with different opinions on specific actions; present their strengths and weaknesses. Better to choose fewer and discuss them.

Response to Comment (2.3): Discussion is enhanced with a major modification by implanting international experiences. First, it was deliberated why urban transformation in European and in general Western cities can be fundamentally different from the Middle Eastern counterparts. Later several successful case studies (in a relatively similar context, i.e. the Middle Eastern cities) were added and analysed, including Tunis, Iraq, Egypt, Morroco, and Uzbekistan. The content then was analysed/compared and lessons learnt were extracted and elaborated in each section without numbering. Besides, the research contributions and recommendations also were added to this section (please refer to Section 6).

Comment (2.4). The Conclusions chapter is too long. Some of the questions/sentences in this chapter are more relevant to the Discussion.

Response to Comment (2.4): Conclusions was enhanced and extra material moved to the discussion (please refer to Section 7).

Reviewer 3 Report

Dear authors,

> This is an interesting article, that deals with a subject that is widely disseminated mainly though Anglo-Saxon literature. This insight about Iran can, therefore, enlighten about a specific geographical context. There is, however, a major concern about the structure of the paper that, in my opinion, clearly undermines the quality of the article. This concerns to how the authors address the empirical section. It is referred that 19 interviews were performed and multi-factorial information is produced. However, when reading the empirical section, I do not find them. Are they what the authors placed between parenthesis? If yes, because it not decipherable anyway, please use a simpler coding. If not, clarify, what are abbreviations. Furthermore, it is not clear how the authors come out with the many sub-sections in section 5; is it the authors option or was it retrieved from some source or interview? Still in section 5, because it is very large, in comparison with remaining text, it is advisable that the text is written in less a descriptive way, which in the current writing style somehow resembles a technical report.

> There is something missing in the background of the changes, i.e., what are the reasons for the change the authors refer as: “transformed under capitalism and modernity”? Are you referring to the economic basis of the country, lifestyles (ex. Moving to the periphery), or other? This is important for the contextualization. The text on line 74 onwards is too scarce.

> Nowadays an important body of literature on inner city transformations is devoted to analyse processes such as gentrification or excessive tourism. Your case study presents us something very different. Can the authors address these differences and explain them? For instance, is it something related with the geographical context or can you be analysing an area that is still on another stage, the decline that precedes the processes that I’ve mentioned earlier. I believe this discussion could insert you article into the international debate on the subject.

> The reference “Habibi, 2010” appears several times and should be numbered as the remaining references.

> Please clarify your sentence on line 151 onwards. It is not clear if the authors are mentioning that top-down regeneration projects are less effective that participatory projects, or if it has something to do with the adaptation of western ideas. Either way, it is a strong statement. Please add some discussion on this.

> Please provide more detailed info about: (i) the source of dataset mentioned in section 4; (ii) the institution of the 19 interviewees.

I hope the authors give an opportunity to modify the text. In my opinion, this change must be substantial. The good thing is that, I believe this in-depth change will allow the text to be way better.

Author Response

Response to Reviewer No.3’s comments:

Comment (3.1): This is an interesting article, that deals with a subject that is widely disseminated mainly though Anglo-Saxon literature. This insight about Iran can, therefore, enlighten about a specific geographical context. There is, however, a major concern about the structure of the paper that, in my opinion, clearly undermines the quality of the article. This concerns to how the authors address the empirical section. It is referred that 19 interviews were performed and multi-factorial information is produced. However, when reading the empirical section, I do not find them. Are they what the authors placed between parenthesis? If yes, because it not decipherable anyway, please use a simpler coding. If not, clarify, what are abbreviations. Furthermore, it is not clear how the authors come out with the many sub-sections in section 5; is it the authors option or was it retrieved from some source or interview? Still in section 5, because it is very large, in comparison with remaining text, it is advisable that the text is written in less a descriptive way, which in the current writing style somehow resembles a technical report.

Response to Comment (3.1): As it is truly mentioned by the third reviewer, the 19 interviewees are referenced in the notes, placed between the parenthesis in the text (e.g. ((D-GC)). The table of abbreviation was missed in the previous version of the paper and now it is added as Appendix B, at the end of the text, including the abbreviated name of the interviewees (to protect the confidentiality), their relevant institutions/organizations and the roles of the interviewees within that organization. As an ethical procedure, all personal data were kept private and confidential.

As mentioned by Reviewer #3, the subsections in section 5 have been extracted from the conducted in-depth interviews as major themes, and then rearranged to form each subsection.  Appendix A was also added, indicating the letter of ethics approval for these referenced interviews (please refer to Appendices A and B).

Comment (3.2): There is something missing in the background of the changes, i.e., what are the reasons for the change the authors refer as: “transformed under capitalism and modernity”? Are you referring to the economic basis of the country, lifestyles (ex. Moving to the periphery), or other? This is important for the contextualization. The text on line 74 onwards is too scarce.

Response to Comment (3.2): In response to the issues raised by the reviewer, kindly note that we have added new paragraphs and new citations, indicating the backgrounds of those changes and highlighted the actual reasons behind the modernity that triggered such dramatic changes to the Introduction (please refer to the Introduction, first paragraph as well as Section 1.1.).

Comment (3.3): Nowadays an important body of literature on inner city transformations is devoted to analyse processes such as gentrification or excessive tourism. Your case study presents us something very different. Can the authors address these differences and explain them? For instance, is it something related with the geographical context or can you be analysing an area that is still on another stage, the decline that precedes the processes that I’ve mentioned earlier. I believe this discussion could insert you article into the international debate on the subject.

Response to Comment (3.3): In order to address this comment, we have added Section 1.1. Geographical context and differences in inner-city transformation in historic Middle-Eastern and European cities to the introduction section. Also, new resources were added to back up the discussion (please refer to Section 1.1.).

Comment (3.4): The reference “Habibi, 2010” appears several times and should be numbered as the remaining references.

Response to Comment (3.4): We have now edited this reference.

Comment (3.5): Please clarify your sentence on line 151 onwards. It is not clear if the authors are mentioning that top-down regeneration projects are less effective that participatory projects, or if it has something to do with the adaptation of western ideas. Either way, it is a strong statement. Please add some discussion on this.

Response to Comment (3.5): In response to this comment, 2 more paragraphs were added to this above-mentioned sentence in section 3.1., to further clarify the discussion as requested in the above comment (please refer to section 3.1.).

Comment (3.6): Please provide more detailed info about: (i) the source of dataset mentioned in section 4; (ii) the institution of the 19 interviewees.

Response to Comment (3.6): Appendix B was added at the end of the article addressing further details about the sources of datasets; including the abbreviated name of interviewees, their organization, and their roles in those organizations. This info was also referenced in section 4. Methods (Please refer to Section 4 and Appendix B).  

I hope the authors give an opportunity to modify the text. In my opinion, this change must be substantial. The good thing is that, I believe this in-depth change will allow the text to be way better.

Taken as a whole, we thank the reviewers for the valuable comments. We have tried our best to address all the issues raised and we hope the reviewers would find the revision acceptable.

Round 2

Reviewer 1 Report

I thank the authors for having followed up on my considerations. The text seems to me much more fluid, and certainly better defined in an international overview. I fully agree with the authors that the dynamics that take place in the historic centers of Middle Eastern cities are fundamentally and deeply different from those in the West due to different political, economic and cultural dynamics.
But I think it is very useful, in the economy of the text, to have framed the problem in the international context and then to have compared the Iranian situation with more similar contexts.
I appreciated the efforts of the authors who, in my opinion, made the text of the article more complete, but above all more understandable for all readers.

Although the structure of the text is still quite articulated, to date, this article essentially represents a good qualitative study in which the problems and dynamics relating to DABs in Historic Iranian Cities have been investigated in a fairly in-depth manner. For this reason, in my opinion, it can be accepted in this form.

Author Response

Response to Reviewer No.1’s comments: We are humbled by Reviewer#1’s generous comments on our work. We thank her/him for having the confidence with our work.

Reviewer 2 Report

Dear Authors,

I am pleased to note the large contribution that you have made to the improvement of the article. It is now really more valuable in terms of content and structure.
After reading this version of the article, I have just one more note.
I believe that the Recommendations should not be in the same chapter as the discussion. Rather, Recommendations are (final) part of the Results. There is no discussion in the recommendations, you present your position based on the research. In the discussion, the author's arguments are as important as the arguments of other researchers who deal with this problem.

Author Response

Response to Reviewer No.2’s comments: We thank the reviewer for this observation.

In response to the latter comment, kindly note that we have made necessary changes to improve the clarity of the discussion section. Recommendation statements have been moved to the conclusion section. Meanwhile, the title of section 6 has been changed to “Discussion” for a better understanding as this section is mainly focused on the interpretation of the significant findings of this study.

Reviewer 3 Report

Dear authors,

In your revised version of the manuscript, you have responded to my previous comments.

Having this in consideration, my recommendation for your article is to accept it. As I said previously, your case study is interesting. I hope new and more in-depth analysis of the studied problem can be addressed in upcoming research.

I'm not an english native, therefore I do not qualify to evaluate your english. Still, there is a small typo in line 460. Instead of "form" I believe to "from".

Author Response

Response to Reviewer No.3’s comments: We thank the reviewer for this observation. We apologise for this grammatical error. We have revised it accordingly. 
